# GUNet: A Graph Convolutional Network United Diffusion Model for Stable and Diversity Pose Generation

## Abstract

Pose skeleton images are an important reference in pose-controllable image generation. In order to enrich the source of skeleton images, recent works have investigated the generation of pose skeletons based on natural language. These methods are based on GANs. However, it remains challenging to perform diverse, structurally correct and aesthetically pleasing human pose skeleton generation with various textual inputs. To address this problem, we propose a framework with **GUNet** as the main model, **PoseDiffusion**. It is the first generative framework based on a diffusion model and also contains a series of variants fine-tuned based on a stable diffusion model. PoseDiffusion demonstrates several desired properties that outperform existing methods. *1) Correct Skeletons.* GUNet, a denoising model of PoseDiffusion, is designed to incorporate graphical convolutional neural networks. It is able to learn the spatial relationships of the human skeleton by introducing skeletal information during the training process. *2) Diversity.* We decouple the key points of the skeleton and characterise them separately, and use cross-attention to introduce textual conditions. Experimental results show that PoseDiffusion outperforms existing SoTA algorithms in terms of stability and diversity of text-driven pose skeleton generation. Qualitative analyses further demonstrate its superiority for controllable generation in Stable Diffusion.

## 1 Introduction

As an important external control condition in controllable image generation, 2D pose skeleton images are critical to the quality of the generated images. For example, ControlNet(Zhang et al., 2023), HumanSD(Ju et al., 2023), GRPose(Yin et al., 2024) and other models used to create controllable photos need to be provided with one or more 2D pose skeleton images for reference. In addition, many pose sequence generation tasks also need to be provided with the first frame of the pose skeleton. However, in order to obtain the pose skeletons, current methods rely mainly on extracting them from existing images using pose detection models, such as DWPose(Yang et al., 2023) and OpenPose(Cao et al., 2017). This limits the diversity and operability of obtainable pose skeletons. In practice, there are also blender-based applications which allow users to adjust and model the pose by dragging and dropping key points. However, these approaches are time-consuming and require a lot of manual effort. They are also difficult to implement an end-to-end process due to the large amount of manual intervention. In order to get pose skeleton images more easily and flexibly, it is necessary to create a framework for generating pose skeletons directly from natural language.

In recent years, many researchers have explored methods for generating 2D human pose skeletons from textual descriptions. Zhang et al. (2021) trained a generative adversarial network capable of generating single-person poses from text. Roy et al. (2022) proposed the DE-PASS, a dataset with detailed labelling of the facial details in poses, and trained a generative adversarial network for generating single-person poses conditioned on natural language on it, which designed a RefineNet to improve the generation of facial gestures. However, they share a common problem, i.e., the generated pose skeletons suffer from misplaced key points and disproportionate bone lengths. Fig. 1 shows several illustrations of erroneous skeletons generated by a GANs-based method(Zhang et al., 2021). Fig. 1(a) shows a disproportionate skeleton where the two arms of the character are not the same length. Fig. 1(b) shows a skeleton with misplaced key points. The key point that should be on the

(a)          (b)          (c)

Figure 1: Illustrations of the wrong skeletons generated by GANs-based method. Fig. 1(a) shows a disproportionate skeleton, and Fig. 1(b)shows a skeleton with misplaced key points, and Fig. 1(c)shows a deformed and twisted skeleton.

feet mistakenly appeared near the shoulders. Fig. 1(c) shows a skeleton in which the right knee is folded far beyond the limits of what humans can do, resulting in a partially deformed right leg.

To address the above challenges, we propose PoseDiffusion, a framework capable of generating 2D single-person pose skeletons from various texts. We propose to introduce a denoising diffusion probabilistic model for tasks of text-driven human pose skeleton generation. Unlike classical text-to-image models that use 3-channel feature maps to represent an RGB image, we propose to represent each key point as a separate feature channel, thus enabling individual prediction of the position of each key point. The above approach can significantly increase the diversity and stability of the generated pose images. In addition, to avoid the model always generating deformed skeletons, PoseDiffusion's GUNet model introduces the key point and skeleton information into the generation process through a GCN. As a result, our model generates correct keypoint locations and appropriate skeleton lengths. Furthermore, in order to verify the effectiveness of introducing the skeleton information, we propose two fine-tuned Stable Diffusion-based variants.

We perform extensive qualitative experiments and quantitative evaluations on popular benchmarks. First, we demonstrate significant improvements in text-driven 2D pose skeleton generation. Second, we show the generation capabilities of two fine-tuned variants. In addition, we discuss additional possibilities of PoseDiffusion for multiplayer pose generation.

In summary, our proposed PoseDiffusion has several desired properties that outperform the prior arts:

- *1) Correct Skeletons.* Benefit from our designed denoising model GUNet, PoseDiffusion can use the connection information of the human skeleton as a reference during the denoising process, so as to generate human pose skeletons with the correct location of key points and appropriate length of the bones.

- *2) Diversity.* Benefit from our bold attempt at the conditional diffusion model, decoupled representation of human posture skeleton, and cross-attention design of multimodal information, PoseDiffusion can achieve higher diversity in generating results.

## 2 RELATED WORKS

### 2.1 POSE OR KEYPOINT-GUIDED TEXT-DRIVEN IMAGE GENERATION

With the advent of Stable Diffusion(Rombach et al., 2022)(SD), work on text-driven image generation has made tremendous progress. To avoid pose distortion of people or objects in the generated images, many works use pose skeletons to guide text-driven image generation. ControlNet(Zhang et al., 2023) incorporates additional inputs such as pose information into the SD, allowing the user to precisely control image details during the generation process. HumanSD(Ju et al., 2023)introduces a specific human pose coding module to improve the generation accuracy of fine-grained pose manipulation tasks. T2I-Adapter(Mou et al., 2024) is an insertion module that combines textual cues with additional visual guidance signals to help the generative model better map complex textual descriptions to image content. GRPose(Yin et al., 2024) uses GNNs to process the relationship of skeleton points, which helps the SD model better understand the reference poses, thus generating images with better pose consistency. All of the above work requires an additional picture of the pose skeleton. Current approaches mainly rely on extracting the pose skeleton from existing images using pose detection

models such as DWPose(Yang et al., 2023) and OpenPose(Cao et al., 2017), etc. This limits the diversity and tractability of the available pose skeletons. To address this problem, we need to delve into its upstream work, synthesising high-quality human pose skeletons from diverse texts, thereby enriching the source of pose skeletons for text-driven image generation.

## 2.2 Generating Human Pose Skeletons from Natural Language

Previous work has done some research on generating human poses from natural language. Zhou et al. (2019) change the pose of a person in a given image based on a textual description. Zhang et al. (2021) trained a generative adversarial network capable of generating complex poses from natural language descriptions. Although they achieved good results in terms of action complexity and versatility, they did not take into account the spatial relationships between different key points of human poses, resulting in the generation of poses that are subject to problems such as deformities or skeletal disproportions. To solve this problem, we propose a new text-driven human pose skeleton generation pipeline based on denoising diffusion probabilistic model(DDPM)(Ho et al., 2020). Where the denoising model is based on U-Net, and introduces a graph neural network to introduce the spatial information of the pose. Benefiting from the properties of the DDPM, our pipeline is able to generate more diverse samples.

## 2.3 Text-conditioned U-Net

U-Net was initially applied to solve the problem of medical image segmentation(Ronneberger et al., 2015; Oktay et al., 2018; Zhou et al., 2018), and more recent work combines U-Net with diffusion processes for image generation(Rombach et al., 2022; Dhariwal & Nichol, 2021; Ju et al., 2023; Zhang et al., 2023). These works achieve state-of-the-art performance on a variety of tasks such as unconditional image generation, text-driven image generation(Rombach et al., 2022), and image-driven image generation, and inspired our work. Text-conditioned U-Net injects textual information into image latent features through a cross-attention mechanism, thus effectively guiding the image content during the generation process. The downsampling aims at gradually compressing the spatial dimensions of the input feature maps to extract higher-level global semantic information, while the upsampling is used to progressively recover the spatial resolution, combining the global semantic information with the detailed features, to generate an output image of the same size as the input image.

## 3 Method

We propose a diffusion model-based framework, PoseDiffusion, for generating diverse and skeletally structurally stable 2D human pose skeletons(text2pose). We first give the problem definition in Sec. 3.1, then we provide a general description of the proposed PoseDiffusion in Sec. 3.2, followed by the diffusion model in Sec. 3.3 and the U-Net based denoising model, GUNet, in Sec. 3.4.

## 3.1 Task Definition

Human pose skeleton $H$ contains a set of heatmaps $h_i$, where $i \in \{1, 2, ..., K\}$, $K$ denotes the number of key points, and $S$ is the size of heatmaps. Heatmaps are modelled with Gaussian distribution centred on the coordinates of the keypoints to obtain a 2-dimensional image, where each pixel represents the probability of occurrence of the key point, with darker colours indicating a higher probability. Fig. 2 shows a set of heatmaps of a pose skeleton with 17 key points. In text2pose, given a set of descriptions $\{text_i\}$ and a set of random noise samples with the same shape as $H$, the denoising model will generate a set of heatmaps from the random noise that match the given descriptions.

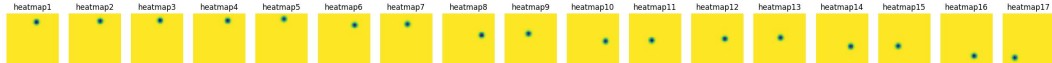

Figure 2: Heatmap of a pose skeleton with 17 key points.

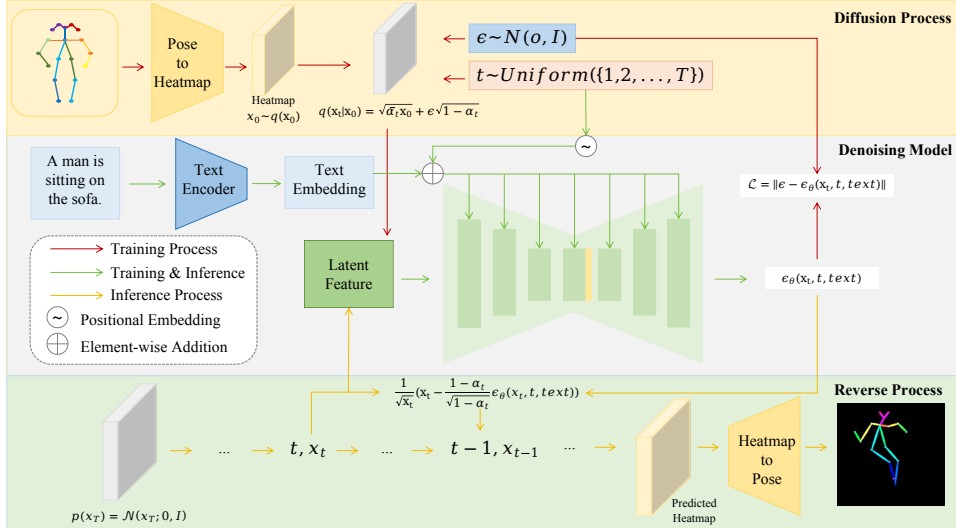

Figure 3: Pipeline overview. In *Diffusion Process*, we transform the pose skeleton into a set of heatmaps via Pose2Heatmap and then add noise to them on timesteps. In the training process, inputs of the *Denoising Model* are noisy latent from *Diffusion Process* and text embedding with timestep embedding, and the output is the predicted noise. *Reverse Process* samples the noise to obtain heatmaps that match the input text and transform them into the pose skeleton via Heatmap2Pose.

## 3.2 PIPELINE OVERVIEW

Our proposed PoseDiffusion pipeline is shown in Fig. 3. First, we construct a text-based pose skeleton generation pipeline using the denoising diffusion model(Nichol & Dhariwal, 2021)(DDPM). The basic of the denoising model is GUNet, a U-UNet-like model. GUNet introduces spatial information of skeletons such as key point locations and connectivity relationships by a spatial module. To enable GUNet to handle textual conditions, we use a cross-attention mechanism to fuse cross-modal features. In addition to this, we change the input dimension of GUNet to match the skeleton features represented using a set of heatmaps. The above design significantly improves the pose skeleton generated by PoseDiffusion. We will describe each part of the pipeline in the following subsections.

## 3.3 DIFFUSION MODEL

The diffusion model is a generative model that progressively denoises Gaussian noise through a learnable probabilistic model. The forward process of diffusion is a Markov chain that starts from the initial data $x_0$ and gradually adds noise with variance $\beta_t$ to the data $x_{t-1}$ over $T$ timesteps to obtain a set of noise samples $x_t$ with distribution $q\left(x_t \mid x_{t-1}\right) = N\left(x_t; \sqrt{1-\beta_t}x_{t-1}, \beta_t I\right)$. The distribution of the whole forward noise addition process can be calculated by Eq. 1:

$$q\left(x_{1:T} \mid x_0\right) = q\left(x_0\right)\prod_{t=1}^{T} q\left(x_t \mid x_{t-1}\right) \tag{1}$$

The inverse process is the reversal of the forward process, where we sample from $q\left(x_t \mid x_{t-1}\right)$ and progressively reconstruct the true sample, which means estimating $q\left(x_{t-1} \mid x_t\right)$ at the moment $t = T$. Estimating the previous state from the current state requires knowledge of all the previous gradients. Thus, it is necessary to train a neural network model to estimate $p_\theta\left(x_{t-1} \mid x_t\right)$ based on the learned weights $\theta$ and the current state at time $t$. This trajectory can be performed by Eq. 2:

$$p_\theta \left( x_{t-1} \mid x_t \right) = N \left( x_{t-1}; \mu_\theta \left( x_t, t \right), \sum_\theta \left( x_t, t \right) \right)$$

$$p_\theta \left( x_{0:T} \right) = p \left( x_T \right) \prod_{t=1}^{T} p_\theta \left( x_{t-1} \mid x_t \right) \tag{2}$$

To provide a simplified representation of the diffusion process, Ho et al. (2020) formulated it as Eq. 3:

$$q \left( x_t \mid x_0 \right) = \sqrt{\bar{\alpha}_t} x_0 + \sqrt{1 - \bar{\alpha}_t}, \epsilon \in N \left( 0, I \right) \tag{3}$$

where $\alpha_t = 1 - \beta_t$ and $\bar{\alpha}_t = \prod_{s=0}^{t} \alpha_s$. Thus we can simply sample the noise samples and generate $x_t$ directly from this formula. Here, we follow the approach used in GLIDE(Nichol et al., 2021) and predict the noise term $\epsilon$. The expression for the neural network prediction during the sampling process can be simplified to $\epsilon_\theta \left( x_t, t, text \right)$. We optimize the denoising model using the loss function as shown in Eq. 4:

$$\mathcal{L} = E_{t \in [1,T], x_0 \sim q(x_0), \epsilon \sim \mathcal{N}(0,I)} \left[ \| \epsilon - \epsilon_\theta \left( x_t, t, text \right) \| \right] \tag{4}$$

To generate samples from a given textual description, we perform noise reduction on the sequence from $p \left( x_T \right) = N \left( x_T; 0, I \right)$. From Equation. 2, we know that we need to estimate $\mu_\theta \left( x_t, t, text \right)$ and $\sum_\theta \left( x_t, t, text \right)$. To simplify this, we set $\sum_\theta \left( x_t, t, text \right)$ to a constant $\beta_t$, and $\mu_\theta \left( x_t, t, text \right)$ is approximated as Eq. 5:

$$\mu_\theta \left( x_t, t, text \right) = \frac{1}{\sqrt{x_t}} \left( x_t - \frac{1 - \alpha_t}{\sqrt{1 - \bar{\alpha}_t}} \epsilon_\theta \left( x_t, t, text \right) \right) \tag{5}$$

### 3.4 GUNet and Components

Since our skeleton generation task requires spatial information of the skeleton during denoising process, it makes CNNs, a structure for processing image data, unable to model the structure of pose skeleton. To solve this problem, Wen et al. (2023) proposed a U-Net-like structure based on GNNs. However, GNNs are unable to perform detailed feature extraction on the representation of the pose skeleton, which is a kind of 2D image data. Therefore, we propose GUNet that combines the advantages of both CNN and GNN, as shown in Fig. 4.

**Text Encoder.** In this paper, we use BERT(Devlin, 2018) to obtain the embedding of the text. Specifically, the sentence $T$ describing a pose skeleton is firstly be spilt by the tokenizer of BERT to get $T_{token} = \{t_1, t_2, .t_L.., \}$. Then, $T_{token}$ is input to the embedding layer of BERT for encoding, using the embedding of CLS token as the sentence representation, i.e., $T \in \mathbb{R}^{1 \times 768}$.

**Pose Encoder and Pose Decoder.** Pose Encoder and Decoder are the Pose2Heatmap and Heatmap2Pose modules in Fig. 3. These are two modules that require no training and enable the transformation between a pose skeleton and a set of heatmaps corresponding to it. Specifically, the COCO dataset comes with keypoint coordinates for each pose skeleton of the class. We generate a heatmap of size $S \times S$ for each keypoints of the pose skeleton, and the value of each pixel point of the heatmap represents the probability that the keypoint occurs here. Assuming that the coordinate of the key point $K_i$ is $(\mu_x, \mu_y)$, then, the pixel value at coordinate $(x, y)$ is computed by Eq. 6

$$heatmap \left( x, y \right) = e^{-\frac{(x - \mu_x)^2 + (y - \mu_y)^2}{2\sigma^2}} \tag{6}$$

Eventually, each pose skeleton is represented as a set of heatmaps by Pose2Heatmap, $H \in \mathbb{R}^{K \times S \times S}$, where $K$ denotes the number of keypoints of a skeleton, and $S$ denotes the size of the heatmap. Heatmap2Pose module directly obtains the coordinates of the largest pixel value on the heatmap as the keypoint coordinate. Finally, the key points are connected and coloured using OpenPose's(Cao et al., 2017) skeleton connection and color rules to get the pose skeleton.

**GUNet.** In GUNet, we define the human pose as a graph structure, notated as $\mathcal{G} = (V, E, A)$, where $V$ represents the set of node features, $E$ represents the set of edges, and $A$ is the adjacency matrix defining the relationships between nodes, defined as Eq. 7:

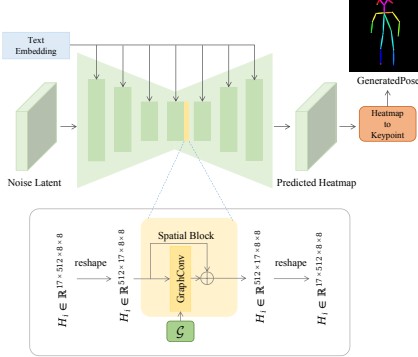

$$A_{ij} = \begin{cases} 1, & \text{if there is an edge between} \\ & \text{node } i \text{ and node } j \\ 0, & \text{others} \end{cases} \quad (7)$$

Figure 4: GUNet. It is a U-Net-like structure consisting of downsampling, upsampling, middle blocks and a spatial block. The sampling and the middle blocks consist of CNN layers, a self-attention layer, and a cross-attention layer. One of the middle blocks is followed by a spatial block containing a graph convolutional neural network layer and a skip connection.

Although the graph structure is fixed for different human pose skeletons, as the pose changes, the associated heatmap sets will be different and the features of the node set $V$ will be updated.

As shown in Fig. 4, GUNet is a U-Net like structure consisting of three down-sampling blocks, three up-sampling blocks, one middle block and one spatial block. Consistent with previous work(Nichol & Dhariwal, 2021), both the downsampling, upsampling and middle blocks consist of CNN layers, and each block contains a self-attention layer and a cross-attention layer for maintaining cross-modal semantic consistency between the textual conditions and the images. We insert a spatial block containing a GCN layer and a skip connection after the middle block. The reason is that the image features include a lot of detailed information at the original resolution.

When the latent embedding $N$ reaches the spatial block of GUNet, we rearrange the dimensions of $N \in \mathbb{R}^{bs \times K_{MID} \times S_{MID} \times S_{MID}}$ to $N \in \mathbb{R}^{K_{MID} \times bs \times S_{MID} \times S_{MID}}$ to facilitate smooth graph convolution computation. The rearranged latent embedding, which serves as the feature for the node set in the graph structure, is then fed into the graph convolution layer. The graph convolution can be formulated as shown in Eq.8.

$$\Gamma_{\mathcal{G}} \left( \bar{N} \right) = \sigma \left( \Phi \left( A_{gcn}, N \right) W \right) \quad (8)$$

where $W \in \mathbb{R}^{bs \times bs}$ denotes the trainable parameters, $\sigma$ is the activation function, and $\Phi \left( \cdot \right)$ is the rule aggregation function that determines how neighbouring features are aggregated into the target node. In this function we directly use the form in the most popular vanilla GCN, defining a symmetric normalised summation function as $\Phi \left( A_{gcn}, N \right) = A_{gcn} N$, where $A_{gcn} = D^{-\frac{1}{2}} \left( A + I \right) D^{-\frac{1}{2}} \in \mathbb{R}^{V \times V}$ is the normalised adjacency matrix of the graph $\mathcal{G}$, $I$ is the unit array, and $D$ is the diagonal matrix, where $D = \left( A + I \right)$. The output of the GraphConv layer is then summed with its input to achieve the skip connection, ensuring the stability of the features. Finally, the positions of $bs$ and $K_{MID}$ in $N \in \mathbb{R}^{K_{MID} \times bs \times S_{MID} \times S_{MID}}$ are exchanged again to produce the output of the spatial module, which is then upsampled by GUNet to generate the predicted heatmap.

## 4 EXPERIMENTS

In this section, we discuss the experimental design and its results. We selected two baseline models based on GAN(Zhang et al., 2021): WGAN-LP Regression (for regression prediction) and WGAN-LP (for heatmap prediction). In addition, we included several baseline models proposed in this paper, including SD1.5-T2Pose(Rombach et al., 2022) with full fine-tuning, PoseAdapter adapted from IPAdapter(Ye et al., 2023), and UNet-T2H, i.e., GUNet without GCNs. We provide a comprehensive comparison of these models with the GUNet proposed in this paper. The dataset, baseline models, and the qualitative and quantitative results are presented separately below.

### 4.1 DATASET & BASELINE MODELS

We use the COCO (Lin et al., 2014) to train and evaluate our models. This dataset contains over 100k annotated images of everyday scenes, each accompanied by five natural language descriptions. Among these, about 16k images feature a single person, with each person annotated by the coordinates of 17 keypoints. To ensure data quality, we filtered approximately 12k single-person images, selecting only those with at least 8 visible keypoints. For each image, one of the five descriptions was randomly selected as the image's label, forming image-text pairs. These pairs were then divided into training and validation sets with a 4:1 ratio.

Our baseline models come from three main sources. First, we fine-tuned Stable Diffusion in two ways: SD1.5-T2P and PoseAdapter, which introduces a pose coding layer. Second, we modified GUNet by removing the graph convolutional neural network layer, resulting in a model we denote as UNet-T2H, where the human posture skeleton is represented as a set of heatmaps. Third, we included two models based on WGAN, focusing on different methods of human posture skeleton representation: WGAN-LP for heatmap prediction and WGAN-LP R for coordinate regression prediction.

The experimental results of the baseline model will be discussed in Sections 4.2 and 4.3.

### 4.2 QUALITATIVE RESULTS

We conducted a series of generative experiments using the above models. We take five text descriptions randomly from the validation set, including common daily actions such as ride, stand, sit, fly a kite, etc., and then compare these generated skeletons with ground truth.

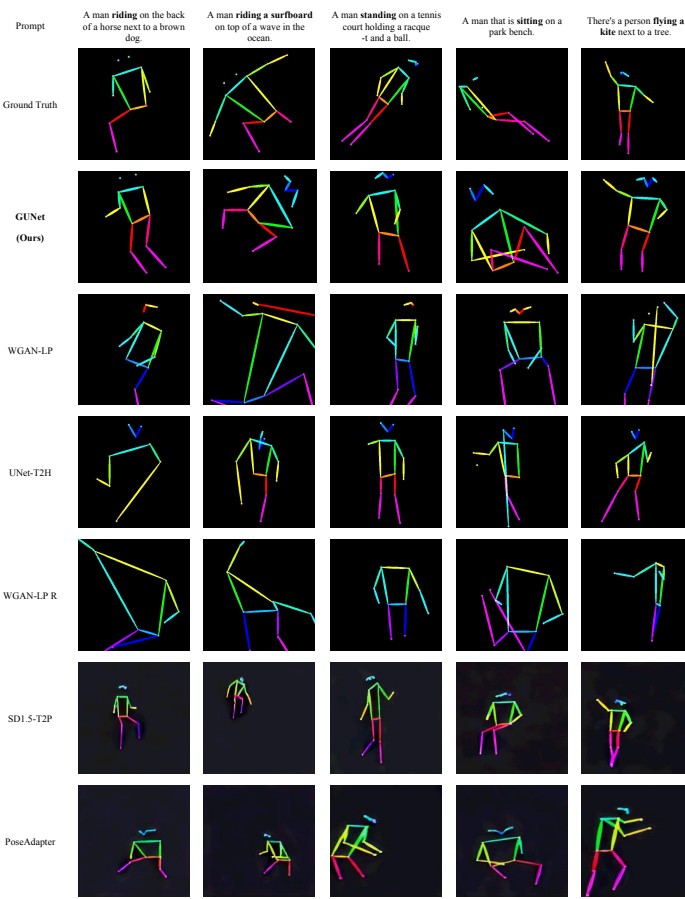

Figure 5: Some qualitative poses generated by the model, using the ground truth pose in the first line as a reference.

The results in Fig.5 show that the human pose skeleton generated by our GUNet (second row of Fig. 5) conforms to the given textual description. And, our results show fewer misplaced keypoints and disproportionate skeletons than other baseline models.

To demonstrate the effectiveness of introducing the spatial relationship of the human skeleton, we compare the results of GUNet (the second row of Fig.5) with those of UNet-T2H (the fourth row of Fig. 5). The UNet-T2H results exhibit clear proportion issues, such as the arms being noticeably longer than normal in the first and second poses of the fourth row, whereas GUNet's results do not have this issue.

To verify the superior performance of the diffusion model compared to GAN, we compare the results of GUNet (the second row of Fig. 5) with those of WGAN-LP (the third row of Fig. 5). It can be seen that the GUNet results are more uniform and coordinated in terms of the distribution of the torso and limbs, such as those generated in the second column, while the poses generated by WGAN-LP are more localised, which is not conducive for subsequent models such as ControlNet to analyse the semantics of the poses.

To evaluate the diversity of human pose skeletons generated by different models, we selected two different scenarios of action descriptions and used each model to generate five human pose skeletons. Since it was demonstrated in the previous paragraph that the model without skeleton connection information significantly underperforms the model with skeleton connection informationn, we simplify the experiments by using only GUNet, WGAN-LP and PoseAdapter for generation.

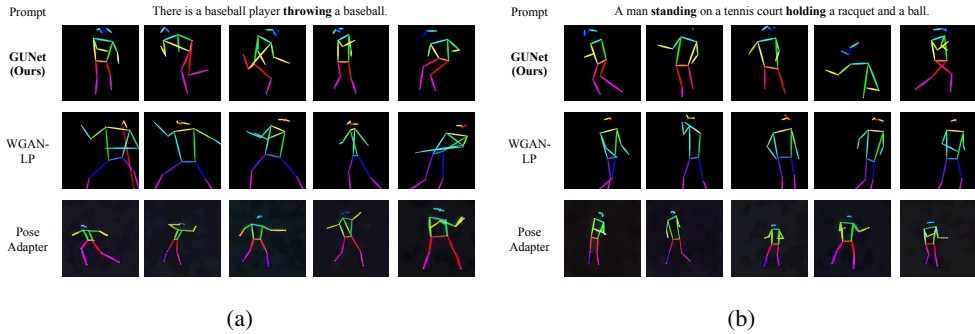

(a)                    (b)

Figure 6: Comparison of the diversity of poses generated by different models.

The results in Fig. 6 show that the pose skeletons generated by WGAN-LP and PoseAdapter exhibit high similarity under the same textual description, i.e., not much of the change in the main skeleton, whereas the pose skeletons generated by GUNet displays greater diversity and variability.

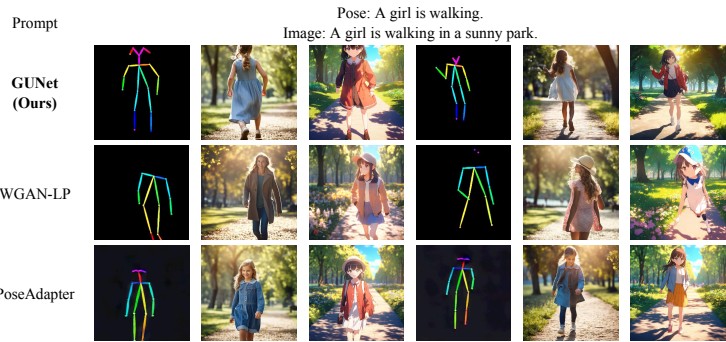

Figure 7: Comparison of the aesthetics of different model-generated poses combined with Controlnet to generate real images.

Finally, to verify the effect of the quality of the human pose skeleton on ControlNet-generated images, we combined pose skeletons generated by different models with ControlNet to generate images via stable diffusion. As can be seen in Fig.7, in WGAN-LP, the poses of the characters often do not match the pose skeletons. It is necessary to point out here that the images do not exactly follow the structure of the pose images that are referenced, as mentioned in Zhang et al. (2023). However, images from WGAN-LP are prone to errors such as multiple arms, which reflects that the pose skeleton generated by WGAN-LP deviates from the real pose distribution and is not coordinated enough as a whole. On the contrary, we introduced the human skeleton information in GUNet and PoseAdapter, which makes the pose of the characters in images generated by ControlNet match well with the skeleton generated by GUNet and PoseAdapter.

## 4.3 QUANTITATIVE RESULTS

The previous section qualitatively analysed the performance of different models by observing the results generated by the models. In this section, we will design a series of experiments for quantitative evaluation to further validate our conclusions.

To validate the accuracy and diversity of model-generated poses, this paper uses MSE and variance as quantitative assessment metrics. Since models based on Stable Diffusion fine-tuning generate pose images directly without first generating intermediate state keypoints, we calculated MSE and variance for four models: GUNet, UNet-T2H, WGAN-LP, and WGAN-LP R. First, we generated 10 pairs of human posture skeletons for the natural language descriptions in the validation set with these models and then compared the coordinates of each keypoint of the generated posture skeletons with the corresponding ground truth to calculate MSE, taking the average value as the accuracy score. Second, we calculated the variance between the coordinates of these 10 pairs of pos-

ture skeletons and took the average value on the validation set as the diversity metric. The experimental results are shown in the first and second columns of Table. 1, where GUNet excels in both MSE and variance metrics, achieving the smallest MSE and the largest variance, representing the highest stability and diversity achieved.

Table 1: Quantitative result 1

| METHODS | MSE | WAR |
|---|---|---|
| **GUNet(Ours)** | **262.2** | **244.5** |
| UNet-T2H | 278.4 | 233.4 |
| WGAN-LP | 286.9 | 172.2 |
| WGAN-LP R | 290.8 | 131.2 |

To compare the aesthetic scores of different model-generated pose skeletons and ControlNet-generated images combined in Stable Diffusion, we employ Hpsv2(Wu et al., 2023) as a quantitative evaluation tool and invite users to make preference choices through a blind selection evaluation. Hpsv2 is a state-of-the-art benchmarking framework for evaluating text-to-image generative models based on the large-scale human preference dataset HPDv2, designed to accurately predict human preferences for generating images.

As shown in Table. 2, we performed Hpsv2 on GUNet, WGAN-LP, and PoseAdapter with scores of 0.2535, 0.2446, and 0.2561, respectively. These results are consistent with the intuition from the qualitative evaluation in Sec.4.2, where the images generated by diffusion-based models have higher aesthetic scores. This further reflects that with the introduction of human pose skeleton information, the generated pose skeletons are closer to real human beings, and perform better in guiding the generation of real-life im-

ages. WGAN-LP, on the other hand, does not introduce human pose information and thus performs poorly in the guidance.

Table 2: Quantitative result 2

| METHODS | HPSV2 | PP | PI |
|---|---|---|---|
| **GUNet(Ours)** | 0.2535 | 43% | 31% |
| WGAN-LP | 0.2446 | 7% | 22% |
| PoseAdapter | **0.2561** | **50%** | **47%** |

In the blind selection evaluation, we designed two experiments for posture skeleton and ControlNet-generated pictures. In pose skeleton part, we generate pose skeletons for a batch of pose description texts and ask the users to choose the one that most closely resembles a real human being, as well as the most aesthetically pleasing skeleton. From the results in the second column of Table. 2, where PP and PI mean user preference for pose skeleton and ControlNet-generated images. PP scores

for different models are 43%, 7% and 50%, respectively. The scores of PoseAdapter and GUNet are much higher than that of WGAN-LP, indicating that the diffusion-based model generates more scientific and aesthetically pleasing pose skeletons from the user' point of view. In ControlNet part, we generate real images using SD combined with ControlNet, and take the pose skeleton generated as the references. Users are asked to choose the images that best match their reference pose skeleton, as well as the ones that are most aesthetically pleasing. As shown in Table. 2, the PI scores are 31%, 22% and 47%, respectively. Although GUNet is rated lower than PoseAdapter, overall, our diffusion-based models far outperform WGAN-LP, which suggests that a scientifically aesthetically pleasing skeleton is important for guiding the effectiveness of controllable image generation.

In addition, by analysing the difference between GUNet and PoseAdapter in terms of user preferences, we conclude that PoseAdapter performs better in terms of compatibility with ControlNet and SD since it is a Stable Diffusion fine-tuned-based model. However, PoseAdapter has a fatal drawback since it treats the pose skeleton as a holistically generated picture, resulting in its generation effect being overly dependent on the quality of the training data, which cannot be further optimised by decoupling the key points when generating misconnections. As a result, although PoseAdapter performs well in ControlNet, the scope for future extensions and optimisations is extremely limited. In contrast, GUNet benefits from the decoupling of key points, which makes it more advantageous for multi-person pose generation and enhanced site-specific pose control tasks.

## 5 CONCLUSIONS

This paper focuses on text-driven 2D pose generation based on the diffusion model, and proposes a PoseDiffusion architecture that includes GUNet, PoseAdapter and so on. GUNet combines UNet with GCN to capture the spatial information of the pose. Extensive experiments and in-depth analyses verify the effectiveness of the proposed framework in text-driven 2D human pose generation. In addition, we also analyse the reasons for the differences in effectiveness presented by GUNet and PoseAdapter, as well as the fact that GUNet's decoupled generation approach holds greater potential in the direction of future multi-person pose generation.

## 6 FUTURE

Future work should further explore the task of multi-person pose generation and detail control. In multi-person pose generation, the inability to distinguish the key points of different individuals leads to unsatisfactory results in generating interactions such as holding hands or hugging. Therefore, the decoupling feature of GUNet can be considered to solve this problem. Meanwhile, the decoupling feature also has great potential for fine-grained control of poses.

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
