# OpenReview forum: "GUNet: A Graph Convolutional Network United Diffusion Model for Stable and Diversity Pose Generation"
_ICLR.cc/2025/Conference — Submitted to ICLR 2025_

### Official Review · Reviewer_rMts · 2024-10-19

**Soundness:** 1
**Presentation:** 3
**Contribution:** 1
**Rating:** 3
**Confidence:** 4

**Summary:**

This paper introduces PoseDiffusion, which uses a diffusion model to generate 2D human pose skeletons from text descriptions. Their denoising model GUNet uses a graph convolutional neural network (GCN) to capture the joint connections within the human skeleton. This results in better-generated poses than those by the Wasserstein GAN with LP normalization.

**Strengths:**

The approach of using a GCN, connecting joints by the human skeleton seems reasonable, although needs to be compared to a pure transformer-based approach.

Considers downstream applications such as using the generated keypoints to control generate images with ControlNet.

**Weaknesses:**

The description of what the baselines are, especially SD1.5-T2P and PoseAdapter, is insufficient to figure out what these models are. It is mentioned that a pose coding layer is added to either just PoseAdapter or both but does not describe well enough what the pose coding layer is and to what part of the model the layer is added.

UNet-T2H is GUNet without the graph convolutional layer so it is an ablation, not a baseline.
The remaining baselines are based on WGAN-LP, a 2018 paper.
This is not to say there are no works to compare against with pose generation. One such work is "Adversarial Synthesis of Human Pose From Text" by Yifei Zhang and colleagues.

Figure 7 and table 2 both claim that a measure of the quality of the generated poses can be measured by the quality of the ControlNet-generated image conditioned on the generated 2d pose. While this is an interesting approach to evaluating generated poses, ControlNet might have its own biases in the images that it can generate that would not help reflect whether or not a conditioned pose is viable. Also, the qualitatives for the ControlNet evaluation do not look convincingly better than the baselines. It would be beneficial to show quantitative improvement, either in FID, Clip-Image similarity or a cycle consistency error on the pose using an off-the-shelf pose detector on the generated images.

I dont see any viable comparisons in this paper and as a result, I cannot recommend this paper as an accept. In order for me to change my review I need to see comparisons against recent work specifically designed to do text to pose estimation. This can include papers that project from 3d to 2d.

**Questions:**

I would like to see some discussion and maybe include a baseline for the benefits of graph convolution over a masked transformer architecture.

I would like to see more evaluations that measure the quality of estimated poses using established evaluation metrics. I would also like to see a better argument for why the ControlNet evaluation proposed in the paper is an effective way of evaluating generated poses, and if so, I would like to see qualitatives on this evaluation.

I would also like to see evaluations against more recent pose generation works.

---

> ### Author Response · Authors · 2024-12-03
>
> Thanks for the suggestions. Here is the explanation of issue.
>
> ## 1) Weaknesses
>
> - The SD1.5-T2P mentioned in the baseline refers to the model obtained by fine-tuning SD1.5 using the Text-Pose pair, where Pose is a 2D human pose skeleton consisting of coloured lines on a black background. The pose encoder mentioned in the PoseAdapter is at the same level as the text encoder in the conditional diffusion model. The essence of the pose encoder is a linear layer, which maps the coding of Pose to 1-dimensional space to get the ‘pose embedding’. The concatenation of the pose embedding and the text embedding is used as a condition for the conditional diffusion model.
>
> - UNet-T2H is indeed performed as an ablation experiment, and in Fig. 5, we show the baseline (WGAN-LP, i.e., the work of Yifei Zhang and colleagues) and the ablation experiment as a whole, in order to more intuitively show the results of the experiment under different conditions.
>
> - In Fig. 7, the second row of WGAN-LP is used as the baseline model,and the final results generated by ControlNet are different from the 2D poses.
>
> ## 2) Problem.
> - UNet-T2H as a GUNet without a graph convolution layer, in the generation results shown in Fig. 5, UNet-T2H generates results with a greater likelihood of malformations and misalignments, whereas GUNet generates results that are relatively correct due to the addition of the constraints of the structure of the human body graph.
>
> - The reason for using ControlNet to evaluate the effectiveness of generating poses is that, ControlNet is the mainstream pose controllable generation method in the industry. However, in fact, our method is not only adapted to ControlNet, but is also general in other networks used for pose controllable generation. For example, we try to use the pose generated by GUNet as a reference in HumanSD (2023), and the generation results are shown in the table below. The final image generated is in the style of shadow puppet theatre, and it can be seen that our GUNet-generated pose can be a good guide for HumanSD to generate the final result in some common everyday actions.
>
>
> |Text|Image|
> |---|---|
> |A man is walking.|[image.png](https://s2.loli.net/2024/12/03/JZoslti4E5O1rFe.png)|
> |A man riding a surfboard on top of a wave in the ocean.|[image.png](https://s2.loli.net/2024/12/03/9WUXtuQYxATZJPw.png)|
> |A girl is sitting on a chair.|[image.png](https://s2.loli.net/2024/12/03/LmoQeVtusF1KvUJ.png)|
> |A woman is cooking a meal.|[image.png](https://s2.loli.net/2024/12/03/MAO58UEh2dTYLzb.png)|
>
> - Recent works on pose generation mainly generate 3D poses from text, such as PoseScript, PoseGPT, etc. These methods generate 3D poses from text. In the main application scenario ‘Controlled Generation of Gestures for 2D Plane’ in our thesis, the above methods have some problems:
>
> #### (1) Although it is possible to convert the results of 3D pose generators to xyz coordinates using models such as smplx, the conversion from xyz to 2D coordinates is often not accurate enough, and there are problems such as overlapping and misalignment of bones. (We tried PCA, TSNE, or direct projection to xy, yz, or xz planes, and could not get better results)
>
> #### (2) The above method focuses on generating fine-grained skeletal poses, such as the position of the left hand, facial orientation, etc. It is not possible to understand the generalised behavioural description. It is not possible to understand generalised behavioural descriptions, such as ‘a person is running’, ‘a person is skiing’, and so on.

---

### Official Review · Reviewer_ehBF · 2024-11-01

**Soundness:** 2
**Presentation:** 1
**Contribution:** 2
**Rating:** 3
**Confidence:** 4

**Summary:**

The paper introduces GUNet, a key component of the PoseDiffusion framework, which generates diverse and structurally stable 2D human pose skeletons from text descriptions. GUNet combines graph convolutional networks with U-Net to capture spatial relationships in human poses, addressing challenges in pose skeleton generation. The framework outperforms existing methods in terms of stability and diversity, offering correct skeletons and enhanced controllability in pose generation.

**Strengths:**

The paper presents GUNet within the PoseDiffusion framework, an original approach that combines graph convolutional networks with U-Net for text-driven 2D human pose skeleton generation, showcasing creativity in integrating spatial information into the model.

**Weaknesses:**

1. The manuscript would benefit from a thorough language revision to enhance clarity and readability. The current phrasing is in need of refinement to better convey the research's contributions and methodology.
2. The presentation of the experimental results, particularly the visualizations, falls short of the standards expected for a top-tier conference. There is a need for more compelling and clear visual representations that effectively communicate the outcomes of the research.
3. The novelty of the proposed model, as it stands, is insufficient to distinguish it from existing work in the field. The manuscript should either provide a more detailed justification for the model's innovative aspects or align its claims with the actual contributions to avoid overstatement.
4. Given the current state of the manuscript, with its language, visualization, and novelty issues, it may not be suitable for publication in a top conference. Considerable improvements are necessary to meet the high standards of such venues.

**Questions:**

Please refer to the weakness part.

---

> ### Author Response · Authors · 2024-12-03
>
> Thanks for the suggestion, we will optimise the additional experiments and related expressions.

---

### Official Review · Reviewer_RooE · 2024-11-03

**Soundness:** 2
**Presentation:** 2
**Contribution:** 2
**Rating:** 3
**Confidence:** 3

**Summary:**

The paper proposes a diffusion model-based framework for the task of generating diverse and skeletally structurally stable 2D human pose skeletons (text2pose). The propose model is called PoseDiffusion, a denoising model that incorporates graphical conv networks and learns the spatial relationships of the human skeleton. During training, it represents 2D skeletons as heatmaps (one per keypoint) and add noise to the heatmap. The denoising process is managed by a model called GUNet—a U-Net-like structure based on graph neural networks (GNNs) with graph convolutional layers—to utilize the graph-like nature of human keypoints.

**Strengths:**

- Novelty. The model takes advantage of the inherent graph structure of human keypoints, allowing it to better capture spatial relationships and dependencies among keypoints. To my knowledge, GNN is not used in human-specific diffusion-based generation yet.

**Weaknesses:**

- Motivation. The introduction suggests that the motivation behind PoseDiffusion is to improve the quality of 2D pose skeleton images for controllable human image generation. However, the experiments section does not provide sufficient evidence to substantiate this motivation, as the quantitative evaluation is performed only at the heatmap level. It remains unclear if improvements at the heatmap level translate to better final image generation.
- Performance. The superiority of this method is not convincing to me through the qualitative results.
  - In the qualitative results (Figure 5) it is unclear whether the proposed method is better than baselines. For example, I think SD 1.5 T2P has more reasonable generation than GUNet.
  - In Figure 7, it seems that the proposed method does not differentiate between left vs right keypoints.
- Evaluation protocol.
  - Line 455 “we generated 10 pairs of human posture skeletons for the natural language descriptions in the validation set with these models and then compared the coordinates of each keypoint of the generated posture skeletons with the corresponding ground truth to calculate MSE”. How can you compare to the GT since this is a generation task and the generations may be different each time?
  - In Table 2 the authors use HPSV2 as a quantitative metric. I am not sure if it’s a valid metric since the paper is generating 2d keypoint heatmaps instead of natural images. The author should explain how they apply this metric in detail.

To conclude, I don't think the paper is ready for publication in its current form. I look forward to hearing other reviewers' insights and perspectives on this work.

**Questions:**

-	In Figure 7, why are the human generations from GUNet sometimes flipped? Shouldn’t the model differentiate left vs. right keypoints  which determines whether the person is facing the camera?

---

> ### Author Response · Authors · 2024-12-03
>
> Thanks for the comments, here is an explanation of what is relevant.
> ## 1) Weaknesses
>
> ### Motivation:
>
> The final image is generated by a pose controllable generation model such as ControlNet, and the main purpose of this study is to provide a high-quality 2D pose skeleton for models such as ControlNet. Other studies, such as pose detection, demonstrated that heatmap-based pose prediction is better than coordinates.
>
> ### Performance:
>
> - As mentioned in the last paragraph of Experiment Section 4.3, although SD1.5 T2P seems to have a more reasonable generation than GUNet, it generates the pose skeleton as a single RGB picture, so it is difficult to optimise the position of different key points further. It does not have an advantage in more complex poses and multi-person poses. Therefore GUNet is superior.
> - Regarding the issue that there is no distinction between left and right keypoints in Fig. 7, the explanation is as follows: the skeleton map distinguishes between left and right according to the different colours of the bones, e.g., blue for the left shoulder and green for the right shoulder, or yellow for the left shoulder and orange for the right shoulder, and so on. However, there is a deviation in the adaptation with ControlNet and other models, resulting in the generated final image not accurately identifying the left and right keypoints.
>
> ### Evaluation:
>
> - Comparison with GT is done to determine whether the model-generated pose matches the input description.
> - HPSV2 measures the aesthetic score of the final image generated by a pose control model such as ControlNet, not the pose skeleton.
>
> ## 2) Question:
>  ‘In Figure 7, why do the human generations in GUNet sometimes flip? Shouldn't the model distinguish between left and right keypoints to determine if the person is facing the camera?’
>
> ### Answer:
> Thanks for the question. the human pose skeleton generated by GUNet distinguishes between left and right keypoints by different line colours, e.g. blue for the left shoulder and green for the right shoulder; or yellow for the left shoulder and orange for the right shoulder. After inputting the human posture skeleton generated by GUNet into ControlNet, the final result sometimes cannot distinguish left and right, which may be due to the lack of adaptability between the result generated by GUNet and ControlNet, etc. We will improve it in the future.

---

### Meta-Review · Area_Chair_LWwK · 2024-12-13

**Metareview:**

The paper addresses the task of generating diverse and structurally stable 2D human pose skeletons from text descriptions using graph convolutional networks to capture spatial relationships in human poses. The reviewers agree that capturing the relation between keypoints using a GCNN is a novel and interesting idea. However, they unanimously agree that the paper is not ready for publication yet, which the main reasons being a lack of comparison to related work and the insufficient quality of the writing and visualizations. The AC agrees with the concerns of the reviewers.

**Additional Comments On Reviewer Discussion:**

The reviewers unanimously agree that the paper is not ready for publication. Several concerns were not sufficiently addressed in the rebuttal.

---

### Decision · Program_Chairs · 2025-01-22

Reject